# The Effect of Feeding Hens a Peanut Skin-Containing Diet on Hen Performance, and Shell Egg Quality and Lipid Chemistry

**Ondulla Toomer [1],\*, Thien Vu [1], Rebecca Wysocky [2], Vera Moraes [2], Ramon Malheiros [2] and Kenneth Anderson [2]**

[1] Food Science & Market Quality Handling Research Unit, ARS, USDA, Raleigh, NC 27695, USA; Thien.Vu@usda.gov

[2] Prestige Department of Poultry Science, North Carolina State University, Raleigh, NC 27695, USA; becca_wysocky@ncsu.edu (R.W.); vmmoraes@ncsu.edu (V.M.); rdmalhei@ncsu.edu (R.M.); ken_anderson@ncsu.edu (K.A.)

\* Correspondence: Ondulla.Toomer@usda.gov

**Abstract:** Peanut skins are a considerable waste product with little current economic value or use. We aimed to determine the dietary effects of peanut skins on layer production performance and egg quality and chemistry of the eggs produced. Two hundred commercial hens were randomly assigned to four treatments (five replicates) and fed ad libitum for 8 weeks: conventional control diet, diet containing 24% high-oleic peanut (HOPN), diet containing 3% peanut skin (PN Skin), and a diet with 2.5% oleic acid (OA). Hens fed the HOPN diet had significantly reduced body weights relative to the control and PN Skin treatments, producing fewer total eggs over the 8-week experimental period. Eggs weights were similar between the control and PN Skin treatments at weeks 2 and 4, while eggs from the PN Skin treatment group were heavier than other treatments at weeks 6 and 8 of the experiment. Eggs produced from the HOPN treatment had reduced saturated fatty acid (FA) content in comparison to the other treatment groups, while similar between PN Skin and control eggs at week 8 of the experiment. This study suggests that PN skins may be a suitable alternative layer feed ingredient.

**Keywords:** alternative layer feed ingredient; peanut skins; high-oleic peanuts; shell eggs; layers; poultry feeding trial

## 1. Introduction

Feed ingredients used to make dietary rations for food production animals account for approximately 60–70% of the total production cost annually [1]. Peanut skins, which are an abundant low-value waste by-product of the peanut industry, contain residual nutrients that may serve as an energy-rich, antioxidant-rich, affordable feed additive or ingredient for production animals. Peanut skins contain 19% fat, 12% fiber, and 14% to 15% polyphenolic compounds [2]. Nevertheless, approximately 70 million pounds of peanut skins are discarded annually with no identified uses and little to no economic value [3].

Dairy and beef cattle feeding trials have shown that peanut skin dietary inclusion rates greater than 8–16% inhibits protein digestion and absorption due to the high content of tannin and procyanidin [4,5]. Interestingly, reduction in the tannin and procyanidin content in peanut skins by ammoniation did not improve protein digestibility, nitrogen retention, or production performance in steers [6]. In contrast, a small goat feeding trial using whole peanuts and/or peanut skins in the diets of goats demonstrated that whole peanuts and/or peanut skins had similar rates of rumen digestibility as conventional forages such as alfalfa hay cubes, while peanut skins providing a high level of dietary antioxidants [7]. While there are several published reports on the use of peanut skins as a feed additive in ruminant diets, there are no published peanut skin feeding trials to date in monogastric production animals. Hence, in this study, we aimed to determine the effect of peanut skins as a feed ingredient on the production performance of layers.

Secondly, we aimed to determine the effect of peanut skins on the chemical composition and quality of the eggs produced from layers fed a peanut skin-containing diet. Poultry feeding trials using carotenoid-rich feed ingredients such as tomato powder, alfalfa concentrate, and marigold extract demonstrate significant enrichment of egg yolk color intensity and carotenoid content in eggs produced from quail [8] and layers [9] versus conventionally fed hens. However, commercial use of these feeding programs is costly and often not viable due to the high cost of inclusion in the diets. Interestingly, studies have shown that hens fed a diet containing peanuts with the skin intact produced eggs enriched in yolk color (2-fold) and in β-carotene content conventional eggs [10]. For this reason, we aimed to determine the effect of feeding peanut skins or oleic acid on egg yolk color and/or chemistry in the eggs produced from hens fed a peanut skin or oleic acid-supplemented diet.

## 2. Materials and Methods

All animal research procedures used in these feeding trials were approved by the North Carolina State University Institutional Animal Care and Use Committee (IACUC #17-001-A).

### 2.1. Experimental Design, Animal Husbandry, Dietary Treatments, and Hen Performance

Two hundred 40-week-of-lay Hy-Line W36 hens were randomly assigned to one of 4 isonitrogenous (18% crude protein) and isocaloric (3080 kcal/kg) treatments, with 5 replicates per treatment, to meet and/or exceed the NRC requirements for layers. Hens were individually housed and fed *ad libitum* for 8 weeks one of the following dietary treatments: control conventional soybean meal + corn, 24% unblanched high-oleic peanut (HOPN), 3% peanut skin (PN Skin), or 2.5% food-grade oleic acid (OA)-supplemented diet. High-oleic peanuts were crushed using a roller mill into crumbles prior to inclusion in the finished HOPN diet. The OA diet was prepared by supplementing the control diet with 2.5% food-grade OA (Millipore Sigma, Burlington, MA, USA). Peanut skins were collected after the blanching process and were ground finely using a blender into a powder prior to inclusion in the diet. There were five replicates per treatment with hens individually housed in battery cages (each cage 12 inches wide × 18 inches deep × 18 inches height) in one room at the Chicken Education Unit, NC State University (Raleigh, NC, USA). Hens were provided feed and water *ad libitum* and 14 L:D 8 weeks. Finished feed samples were analyzed for aflatoxin and microbiological contaminants by the NC Department of Agriculture and Consumer Services, Food and Drug Protection Division Laboratory (Raleigh, NC, USA).

Body weights were recorded for each individual hen at week 1 and week 8, with feed weights recorded weekly. Shell eggs were collected, enumerated, and weighed daily. The total number of eggs produced per treatment was calculated for each experimental week and for the total 8-week feeding trial. The average feed conversion ratio (FCR) was calculated as total feed consumed over the 8-week feeding (kg)/dozens of eggs produced for each treatment group over the 8-week feeding trial.

### 2.2. Egg Quality and Grading

Bi-weekly DSM egg yolk color, Haugh unit (HU), albumen height, vitelline membrane strength, and USDA grade were determined bi-weekly with 15 eggs per treatment by randomly selecting 3 eggs from each replicate. Fresh shell eggs were collected on the day of quality assessment and USDA grading. Haugh unit values were determined using methods described by Haugh [11] and were recorded with the Technical Services and Supplies (TSS) QCD system (Dunnington, York, UK). The QCD system was calibrated to the DSM Color Fan consisting of a series of 15 colored plastic tabs with a range of yolk colors from light yellow to orange-red (color index 1 to 15) defined by Vuillemier [12]. In general, a texture analyzer (TA.XTplus, Stable Micro Systems Ltd., Surrey, UK) was used to measure the shell strength and vitelline membrane strength by the breaking strength using

a 5 kg load cell per the manufacturer's instructions (Stable Micro Systems Ltd., Surrey, UK) with measurements in grams of force. Vitelline membrane strength was determined using methods described by Jones et al. 2005 with a 2 mm/s test speed and 0.0001 kg trigger force [13]. Modified methods of Jones et al. 2002 were used to measure shell strength with a 2 mm/second test speed and a 0.001 kg trigger force [14].

### 2.3. β-Carotene, Lipid Content, and Fatty Acid Analysis

All experimental diets and eggs were analyzed for total cholesterol, crude fat, fatty acid, and β-carotene content in triplicate by an AOAC-certified lab, ATC Scientific (Little Rock, AK, USA), using AOAC-approved standard chemistry methods. Each egg sample was mixed for homogeneity in a whirl-pak® (Millipore Sigma, St. Louis, MO, USA) bag for 3 min using a Smasher™ Lab Blender (Weber Scientific, Hamilton, NJ, USA). Subsequently, all egg samples were frozen at −20 °C and stored frozen until chemical analysis within two weeks of collection. Frozen homogenous egg samples were shipped on dry ice overnight to the vendor for analysis within 2 weeks of collection. Lipid (total cholesterol, crude fat) and fatty acid analysis of homogenous egg samples and feed samples were analyzed using direct methylation methods as described by Toomer et al. [10]. Total cholesterol was measured as mg cholesterol/100 g sample weight (feed or egg), while crude fat was measured as percentage of gram crude fat/gram sample weight (feed or egg). Fatty acid content was measured as percentage of gram of fatty acid/gram total lipid content of the sample (feed or egg). Methods used to determine β-carotene content in eggs are detailed in the AOAC 958.05 [15] color of egg yolk method. Egg fat hydrolysis methods were determined using the AOAC method 954.02 [16]. Gross energy analysis of feed samples was performed by ATC Scientific using an adiabatic oxygen bomb calorimeter with standard methods.

### 2.4. Statistical Analysis

Each hen served as the experimental unit for all performance data. All performance data were evaluated for significance by one-way analysis of variance (ANOVA) at a significance level of $p < 0.05$ using SAS statistical software (version 9.4). If ANOVA results were significant ($p < 0.05$), Tukey's multiple comparisons t-test was conducted to compare the mean of each treatment group with the mean of every other treatment at $p < 0.05$ significance level. Comparisons were made between body weights (50 birds/treatment), feed intake (50 birds/treatment), feed conversion ratio (50 birds/treatment), and egg weights (total # of eggs collected over the 8-week feeding trial).

Fifteen eggs per treatment (3 eggs per replicate randomly selected) were statistically analyzed by one-way ANOVA ($p < 0.05$) using SAS. Means were separated by least-squares means with Tukey–Kramer adjustment for multiple comparisons ($p < 0.05$) for treatment differences in egg quality parameters (Haugh unit, vitelline membrane strength, shell strength, yolk color score), egg β-carotene content, and egg lipid content (crude fat, total cholesterol, fatty acid content) with 60 total egg samples at each time point (0 week, 2 week, 4 week, 6 week, 8 week). Means were separated by least-squares means with Tukey–Kramer adjustment for multiple comparisons ($p < 0.05$) for treatment differences.

## 3. Results

### 3.1. Feed Analysis

Four experimental diets were formulated (Table 1) to be isocaloric (3080 kcal/kg) and isonitrogenous (18% crude protein). In addition, chemical analysis was performed to determine the crude protein, crude fat, gross energy, and fatty acid profile of the experimental diets (Table 2). As expected, the HOPN dietary treatment had the highest level of oleic fatty acid content relative to the other treatment groups, while the control, PN Skin, and OA dietary treatment groups had the highest levels of linoleic fatty acid content relative to the HOPN dietary treatment (Table 2). Interestingly, the PN Skin dietary

treatment had the greatest percent of omega 3 fatty acid content (Table 2) relative to the other treatment groups.

**Table 1.** Feed formulation of experimental laying hen diets.

| Treatments [1] | | | | |
|---|---|---|---|---|
| | **Control** | **HOPN** | **PN Skin** | **OA** |
| Ingredients | | % (by weight) | | |
| Soybean Meal | 20.4 | 0 | 12.0 | 10.0 |
| Corn | 47.5 | 36.9 | 56.9 | 57.0 |
| High-Oleic Peanut [2] | 0 | 24.0 | 0 | 0.0 |
| Soybean Oil | 7.8 | 0 | 4.4 | 0.0 |
| Wheat Bran | 6.0 | 20.0 | 5.0 | 8.7 |
| Soy Protein Isolate | 5.0 | 5.5 | 7.5 | 7.8 |
| Peanut Skin | 0 | 0 | 3.0 | 0 |
| Oleic Acid Oil | 0.0 | 0.0 | 0.0 | 2.5 |
| Calcium Carbonate | 10.8 | 10.8 | 9.1 | 11.3 |
| Dicalcium Phosphorus | 1.5 | 1.2 | 1.6 | 1.5 |
| Sodium Chloride | 0.3 | 0.3 | 0.3 | 0.3 |
| L-Lysine | 0 | 0.5 | 0.1 | 0.2 |
| DL-Methionine | 0.2 | 0.3 | 0.3 | 0.2 |
| L-Tryptophan | 0 | 0 | 0 | 0 |
| L-Threonine | 0 | 0.1 | 0 | 0 |
| Choline Chloride | 0.2 | 0.2 | 0.2 | 0.2 |
| [3] Santoquin® | 0.1 | 0.1 | 0.1 | 0.1 |
| Mineral Premix [4] | 0.2 | 0.2 | 0.2 | 0.1 |
| Vitamin Premix [5] | 0.1 | 0.1 | 0.1 | 0.1 |
| Selenium Premix [6] | 0.1 | 0.1 | 0.1 | 0.1 |
| Metabolizable Energy [7] | 3080 | 3080 | 3080 | 3080 |

[1] Four isocaloric, isonitrogenous (18% protein) diets were fed to Hy-Line W36 hens for 8 weeks. [2] Treatments: control = conventional soybean meal and corn mash diet, HOPN = (24%) unblanched high-oleic peanut crumbles and corn mash diet, PN Skin = control diet supplemented with 3.0% ground peanut skins, OA = control diet supplemented with 2.5% food-grade oleic fatty acid oil. [3] High-oleic peanuts = unblanched raw whole high-oleic peanut crumbles. [3] Santoquin® = Feed antioxidant and preservative to prevent fat oxidation in stored feed (Novus International, St. Charles, MO, USA). [4] Mineral premix manufactured by NCSU FeedMill, supplied the following per kg of diet: manganese, 120 mg; zinc, 120 mg; iron, 80 mg; copper, 10 mg; iodine, 2.5 mg; and cobalt. [5] Vitamin premix manufactured by NCSU FeedMill supplied the following per kg of diet: 13,200 IU vitamin A, 4000 IU vitamin D3, 33 IU vitamin E, 0.02 mg vitamin B12, 0.13 mg biotin, 2 mg menadione (K3), 2 mg thiamine, 6.6 mg riboflavin, 11 mg d-pantothenic acid, 4 mg vitamin B6, 55 mg niacin, and 1.1 mg folic acid. [6] Selenium premix manufactured by NCSU FeedMill = 1 mg selenium premix provided 0.2 mg Se (as $Na_2SeO_3$) per kg of diet. [7] Metabolizable energy = kcal/kg feed.

**Table 2.** Chemical analysis of experimental laying hen diets.

| Treatments [1] | | | | |
|---|---|---|---|---|
| | **Control** | **HOPN** | **PNSkin** | **OA** |
| Component | % (by weight) | | | |
| Crude Fat [2] | 8.4 | 13.9 | 8.7 | 5.1 |
| Crude Protein | 19.4 | 18.5 | 20.2 | 19.0 |
| Fiber | 2.3 | 3.2 | 1.9 | 2.4 |
| * Palmitic | 10.8 | 6.7 | 10.2 | 10.8 |
| * Steric | 3.8 | 3.2 | 3.6 | 2.7 |
| * Oleic | 22.6 | 74.3 | 27.8 | 25.9 |
| * Elaidic | 1.3 | 0.7 | 1.2 | 1.0 |
| * Linoleic | 52.5 | 7.1 | 48.4 | 45.8 |
| * Omega 3 | 6.618 | 0.1 | 58.5 | 3.2 |
| * Omega 6 | 53.2 | 1.4 | 49.4 | 47.8 |
| Gross Energy [3] | 3506 | 3757 | 3308 | 3085 |

[1] Treatments: control = conventional soybean meal and corn mash diet, HOPN = unblanched high-oleic peanut crumbles (24%) and corn mash diet, PN Skin = control diet supplemented with 3.0% ground peanut skins, OA = control diet supplemented with 2.5% food-grade oleic fatty acid oil. Lipid (crude fat, total cholesterol, fatty acid) and beta-carotene analysis was performed by an AOAC-certified lab, ATC Scientific (Little Rock, AR, USA), using AOAC-approved standard methods. [2] Crude fat content = g crude fat/g total sample weight * 100, * fatty acid content = g of fatty acid/g total lipid * 100. Each value represents the mean ± the standard error for each triplicate sample. [3] Gross energy = kcal/kg feed.

### 3.2. Hen Performance and Egg Weights

Hens fed the OA diet had body weights that were significantly less than the body weights of hens fed the control and PN Skin diets ($p < 0.05$), while body weights were similar between the HOPN and OA dietary treatments at week 1 (Table 3). At week 8, hens fed the HOPN diet had significantly smaller body weights relative to the body weights of hens fed the control and PN Skin ($p < 0.05$) diets, while body weights were similar between hens fed the HOPN and OA diets. In addition, hens fed the HOPN experimental diet had significantly reduced feed intake ($p < 0.001$) and fewer dozens of eggs produced ($p < 0.05$) in comparison to the other treatment groups (Table 3). Nevertheless, there were no significant treatment differences in feed conversion ratio over the 8-week feeding trial.

The weekly average egg weights (Table 4) were the smallest in eggs produced from hens fed the HOPN diet relative to the other treatment groups at week 1 of the feeding trial ($p < 0.05$). At week 2, week 4, week 6, and week 8, egg weights from hens fed the HOPN and OA experiment diet were significantly smaller than eggs produced from hens fed the control and PN Skin experimental diets, while egg weights produced from hens fed the OA experimental diet were significantly greater than eggs produced from hens fed the HOPN diet (week 1, week 2, week 4, and week 8; $p < 0.0001$). Egg weights were similar between the control and PN Skin treatment groups at week 1, week 2, and week 4, while egg weights were significantly higher in the PN Skin treatment group at week 6 and week 8 relative to the controls (Table 4).

**Table 3.** Performance of hens fed an unblanched high-oleic peanut or peanut skin diet and housed in battery cages.

| | Body Weights (kg) | FCR [2] | Feed Intake | Dozen Eggs (kg) | Produced |
|---|---|---|---|---|---|
| Treatment [1] | Week 1 | Week 8 | (kg feed/) | Total for 8 weeks | Dozen eggs |
| Control | 1.6 ± 0.03 [a] | 1.6 ± 0.4 [a] | 2.3 ± 0.05 | 9.5 ± 0.2 [a] | 21.1 ± 1.1 [a] |
| HOPN | 1.5 ± 0.03 [ab] | 1.5 ± 0.4 [b] | 2.3 ± 0.05 | 8.5 ± 0.2 [b] | 17.5 ± 1.1 [b] |
| PN Skin | 1.5 ± 0.03 [a] | 1.6 ± 0.4 [a] | 2.3 ± 0.05 | 9.3 ± 0.2 [a] | 20.2 ± 1.1 [a] |
| OA | 1.5 ± 0.03 [b] | 1.5 ± 0.4 [b] | 2.3 ± 0.05 | 9.5 ± 0.2 [a] | 21.6 ± 1.1 [a] |
| *p*-value * | 0.04 | 0.03 | 0.80 | 0.0002 | 0.01 |

Two hundred Hy-Line W36 hens (40 week of lay) were assigned to one of 4 isonitrogenous (18% crude protein) and isocaloric (3080 kcal/kg) treatments (5 replicates per treatment) and fed 8 weeks *ad libitum*. Body weights were collected at week 1 and week 8 of the study. [1] Treatments: control = conventional soybean meal and corn mash diet, HOPN = 24% unblanched high-oleic peanut crumbles and corn mash diet, PN Skin = control diet supplemented with 3.0% ground peanut skins, OA = control diet supplemented with 2.5% food-grade oleic fatty acid oil. [2] Feed conversion ratio (FCR) = kg total feed intake over the 8-week/total dozen eggs produced over 8 weeks for each treatment group. Each value (body weights and feed intake) represents the mean ± the standard error. [a,b] Means within the same column lacking a common superscript differ significantly ($p < 0.05$). * *p*-value = differences determined by ANOVA $p < 0.05$.

**Table 4.** Egg weights from hens fed an unblanched high-oleic peanut or peanut skin diet and housed in battery cages.

| | Weekly Egg Weights [1] (g) | | | | |
|---|---|---|---|---|---|
| Treatment [2] | Week 1 | Week 2 | Week 4 | Week 6 | Week 8 |
| Control | 59.5 ± 0.5 [a] | 60.5 ± 0.4 [a] | 60.3 ± 0.4 [a] | 60.9 ± 0.4 [b] | 60.4 ± 0.4 [b] |
| HO PN | 58.3 ± 0.5 [b] | 58.4 ± 0.4 [c] | 58.2 ± 0.4 [c] | 59.3 ± 0.4 [c] | 58.3 ± 0.4 [d] |
| PN Skin | 60.1 ± 0.5 [a] | 60.8 ± 0.4 [a] | 60.7 ± 0.4 [a] | 61.8 ± 0.4 [a] | 61.9 ± 0.4 [a] |
| OA | 59.6 ± 0.5 [a] | 59.6 ± 0.4 [b] | 59.5 ± 0.4 [b] | 59.7 ± 0.4 [c] | 59.2 ± 0.4 [c] |
| *p*-value * | 0.02 | <0.0001 | <0.0001 | <0.0001 | <0.0001 |

Two hundred 40-week of lay Hy-Line W36 hens were assigned to one of 4 isonitrogenous (18% crude protein) and isocaloric (3080 kcal/kg) treatments (5 replicates per treatment) and fed 8 weeks *ad libitum*. Body weights were collected at week 1 and week 8 of the study. [1] Weights (g) of eggs were determined daily and weekly for each treatment group. Data represent the weekly (1, 2, 4, 6 and 8 weeks) averages ± standard error for each time point for each treatment group. [2] Treatments: control = conventional soybean meal and corn mash diet, HOPN = 24% unblanched high-oleic peanut crumbles and corn mash diet, PN Skin = control diet supplemented with 3.0% ground peanut skins, OA = control diet supplemented with 2.5% food-grade oleic fatty acid oil. Each value represents the mean ± the standard error. [a,b,c,d] Means within the same column lacking a common superscript differ significantly ($p < 0.05$). * *p*-value = differences determined by ANOVA $p < 0.05$.

### 3.3. Egg Grading and Quality

All eggs produced in this 8-week feeding trial were graded as USDA Grade AA of superior quality, with thick, firm egg whites and defect-free egg yolks. Moreover, all shells were clean and without defects. There were a minimal number of blood spots or number of meat spots and no statistical difference at the 95% confidence interval between eggs produced from the treatment groups (data not shown). There were no significant differences in 8-week average shell strength or vitelline membrane strength between shell eggs produced from hens fed the four different treatments (Table 5). However, the HU used as a measurement of egg quality was similar between shell eggs produced from hens fed the control, HOPN, and PN Skin dietary treatments, while the 8-week average HU of eggs produced from hens fed the OA diet was significantly lower than shell eggs from the HOPN and PN Skin treatment groups ($p < 0.05$). Of most interest, the 8-week average yolk color was significantly less in eggs produced from hens fed the HOPN experimental diet in comparison to the other treatment groups ($p < 0.0001$).

**Table 5.** Egg quality of eggs produced from hens fed an unblanched high-oleic peanut or peanut skin diet and housed in battery cages.

| . | Shell Strength | Vitelline Membrane | Haugh Unit | Yolk Color |
|---|---|---|---|---|
| | (g force) | Strength (g force) | (HU) | Roche (1–15) [1] |
| Treatment [2] | | Weekly Average (8-Week Study) | | |
| Control | 3742 ± 209 | 0.2 ± 0.007 | 83.3 ± 1.7 [ab] | 3.0 ± 0.2 [a] |
| HO PN | 3828 ± 209 | 0.2 ± 0.007 | 86.1 ± 1.7 [a] | 1.8 ± 0.2 [b] |
| PN Skin | 3770 ± 209 | 0.2 ± 0.007 | 85.4 ± 1.7 [a] | 2.9 ± 0.2 [a] |
| OA | 3979 ± 209 | 0.2 ± 0.007 | 81.7 ± 1.7 [b] | 2.9 ± 0.2 [a] |
| *p*-value * | 0.68 | 0.31 | 0.04 | <0.0001 |

Two hundred Hy-Line W36 hens (40-week of lay) were assigned to one of 4 isonitrogenous (18% crude protein) and isocaloric (3080 kcal/kg) treatments (5 replicates per treatment) and fed 8 weeks *ad libitum*. Eggs were collected on the day of quality assessment with 15 eggs per treatment, with 3 eggs randomly selected per replicate. Each value represents the average values over the 8-week period ± SEM. [1] Yolk color = Roche Color Fan color index 1–15 (lightest to darkest color intensity). [2] Treatments: control = conventional soybean meal and corn mash diet, HOPN = 24% unblanched high-oleic peanut crumbles and corn mash diet, PN Skin = control diet supplemented with 3.0% ground peanut skins, OA = control diet supplemented with 2.5% food-grade oleic fatty acid oil. * *p*-value = differences determined by ANOVA *p* < 0.05. [a,b] is described as items within a column sharing the same superscript are similar, so that means that Control and OA are similar statistically, while HOPN, PN Skin and Control are statistically similar, but OA and HOPN and PN Skin are statistically different.

### 3.4. Egg Chemistry

There were no significant treatment differences in total cholesterol and crude fat (CF) levels in eggs produced from hens fed the four dietary treatment groups at any of the time points measured (Table 6). Eggs produced from hens fed the HOPN diet had the lowest content of saturated fatty acid levels of palmitic and stearic acid, in comparison to eggs produced from the other dietary treatment groups at week 2, week 4, week 6, and week 8 (*p* < 0.0001, Table 6). In contrast, eggs produced from hens fed the OA dietary treatment had the highest content of palmitic saturated fatty acid levels in comparison to eggs produced from the other dietary treatment groups at week 2, week 4, and week 6 (*p* < 0.0001). Palmitic fatty acid content was similar between eggs produced from hens fed the control diet and PN Skin diet at week 4, week 6, and week 8 of the experimental timeframe. Moreover, eggs produced from hens fed the OA diet had significantly reduced levels of stearic saturated fatty acid levels in comparison to eggs produced from hens fed the control diet and PN Skin diet at week 2, week 4, week 6, and week 8 (*p* < 0.0001). Stearic saturated fatty acid levels were similar between eggs produced from hens fed the control diet and PN Skin diet at week 2, week 4, and week 8 (*p* < 0.0001).

Eggs produced from hens fed the HOPN dietary treatment had the lowest level of trans-fat elaidic acid in comparison to eggs produced from hens fed the other dietary treatments at week 4, week 6, and week 8 (*p* < 0.0001, Table 6). However, eggs produced from hens fed the control diet and the PN Skin diet had similar levels of elaidic fatty acid content at week 4, week 6, and week 8. Ironically, at week 2, eggs produced from hens fed the HOPN treatment had the highest content of elaidic acid content compared to eggs produced from the other treatment groups. Oleic fatty acid content was highest in eggs produced from hens fed the HOPN experimental diet, followed by eggs produced from hens fed the OA and PN Skin experimental diets at week 2, week 4, week 6, and week 8 (*p* < 0.0001, Table 6). Eggs produced from hens fed the control diet had the lowest levels of oleic acid content relative to eggs produced from hens fed the other dietary treatments at all time points measured.

**Table 6.** Lipid and fatty acid content of eggs produced from hens fed unblanched high-oleic peanut or peanut skins and housed in battery cages.

| Week | Trmt [1] | Cholesterol | CF | Palmitic | Stearic | Elaidic | Oleic |
|---|---|---|---|---|---|---|---|
| 2 | Control | N/A | 4.7 ± 0.9 | 23.1 ± 0.3 [b] | 8.8 ± 0.1 [a] | 30.7 ± 0.4 [d] | 30.7 ± 0.4 [d] |
| | HO PN | N/A | 5.5 ± 0.9 | 17.8 ± 0.3 [c] | 6.0 ± 0.1 [c] | 62.3 ± 0.4 [a] | 62.3 ± 0.4 [a] |
| | PN Skin | N/A | 5.7 ± 0.9 | 24.2 ± 0.3 [a] | 8.9 ± 0.1 [a] | 35.6 ± 0.4 [c] | 35.5 ± 0.4 [c] |
| | OA | N/A | 4.7 ± 0.9 | 24.9 ± 0.3 [a] | 7.5 ± 0.1 [b] | 44.9 ± 0.4 [b] | 44.9 ± 0.4 [b] |
| | *p*-value * | N/A | 0.63 | 0.001 | <0.0001 | <0.0001 | <0.0001 |
| 4 | Control | 255 ± 41 | 6.4 ± 0.8 | 24.1 ± 0.2 [b] | 9.2 ± 0.2 [a] | 1.4 ± 0.1 [b] | 32.8 ± 0.3 [d] |
| | HO PN | 240 ± 41 | 5.6 ± 0.8 | 17.9 ± 0.2 [c] | 5.9 ± 0.2 [c] | 0.9 ± 0.1 [c] | 63.9 ± 0.3 [a] |
| | PN Skin | 211 ± 41 | 4.9 ± 0.8 | 24.0 ± 0.2 [b] | 8.8 ± 0.2 [a] | 1.3 ± 0.1 [b] | 36.6 ± 0.3 [c] |
| | OA | 229 ±41 | 6.2 ± 0.8 | 25.4 ± 0.2 [a] | 7.5 ± 0.2 [b] | 1.6 ± 0.1 [a] | 46.3 ± 0.3 [b] |
| | *p*-value * | 0.75 | 0.30 | <0.0001 | <0.0001 | 0.0002 | <0.0001 |
| 6 | Control | 261 ± 35 | 5.7 ± 0.8 | 22.9 ± 0.4 [b] | 29.9 ± 0.3 [a] | 1.3 ± 0.05 [b] | 30.0 ± 0.8 [d] |
| | HO PN | 296 ± 35 | 6.5 ± 0.8 | 17.1 ± 0.4 [c] | 5.6 ± 0.3 [d] | 0.9 ± 0.05 [c] | 58.3 ± 0.8 [a] |
| | PN Skin | 248 ± 35 | 5.7 ± 0.8 | 23.5 ± 0.4 [b] | 8.4 ± 0.3 [b] | 1.3 ± 0.05 [b] | 34.5 ± 0.8 [c] |
| | OA | 291 ± 35 | 6.8 ± 0.8 | 24.4 ± 0.4 [a] | 7.4 ± 0.3 [c] | 1.6 ± 0.05 [a] | 43.4 ± 0.8 [b] |
| | *p*-value * | 0.50 | 0.40 | <0.0001 | <0.0001 | <0.0001 | <0.0001 |
| 8 | Control | 312 ± 22 | 9.6 ± 0.2 | 23.3 ± 0.4 [a] | 8.9 ± 0.2 [a] | 1.2 ± 0.05 [b] | 30.7 ± 0.8 [c] |
| | HO PN | 298 ± 22 | 9.8 ± 0.2 | 18.1 ± 0.4 [b] | 6.2 ± 0.2 [c] | 0.9 ± 0.05 [c] | 58.6 ± 0.8 [a] |
| | PN Skin | 281 ± 22 | 9.3 ± 0.2 | 22.8 ± 0.4 [a] | 8.5 ± 0.2 [a] | 1.2 ± 0.05 [b] | 42.7 ± 0.8 [b] |
| | OA | 279 ± 22 | 9.6 ± 0.2 | 23.3 ± 0.4 [a] | 7.3 ± 0.2 [b] | 1.4 ± 0.05 [a] | 41.2 ± 0.8 [b] |
| | *p*-value * | 0.40 | 0.08 | <0.0001 | <0.0001 | <0.0001 | <0.0001 |

Two hundred Hy-Line W36 hens (40-week of lay) were assigned to one of 4 isonitrogenous (18% crude protein) and isocaloric (3080 kcal/kg) treatments (5 replicates per treatment) and fed 8 weeks *ad libitum*. Eggs were collected weekly, and 15 eggs/treatment (3 eggs randomly selected/replicate) N = 60 were chemically analyzed bi-weekly at an AOAC-certified commercial lab, ATC Scientific (Little Rock, AR, USA), using standard AOAC-approved methods. N/A = total cholesterol could not be analyzed at week 2 of the study due to lack of sample volume. Each value represents the average values over the bi-weekly period ± SEM. [1] Treatments: control = conventional soybean meal and corn mash diet, HOPN = unblanched high-oleic peanut crumbles (24%) and corn mash diet, PN Skin = control diet supplemented with 3.0% ground peanut skins, OA = control diet supplemented with 2.5% food-grade oleic fatty acid oil. * *p*-value = differences determined by ANOVA. [a,b,c,d] Means within the same column lacking a common superscript differ significantly (*p* < 0.05).

Omega 3, omega 6, linoleic and linolenic fatty acid content was the lowest in eggs produced from hens fed the HOPN experimental diet, followed by eggs produced from hens fed the OA experimental diet, relative to the other dietary treatment groups at week 2, week 4, week 6, and week 8, with the exception of omega 3 content at week 6 (*p* < 0.0001, Table 7). Eggs produced from hens fed the control diet had the highest levels of omega 3, omega 6, linoleic and linolenic fatty acid content, followed by eggs produced by hens fed the PN Skin dietary treatment relative to eggs produced from hens fed the other dietary treatments at week 2, week 4, week 6 and week 8. There were no significant treatment differences in β-Carotene content in eggs produced from hens fed the four dietary treatments at any time point measured.

**Table 7.** β-Carotene and fatty acid content of eggs produced from hens fed unblanched high-oleic peanut or peanut skins and housed in battery cages.

| | Week | Omega 3 | Omega 6 | Linoleic | Linolenic | β-Carotene |
|---|---|---|---|---|---|---|
| 2 | Control | 1.8 ± 0.02 [a] | 30.5 ± 0.5 [a] | 24.9 ± 0.2 [a] | 1.6 ± 0.02 [a] | 3.9 ± 0.6 |
| | HO PN | 0.3 ± 0.02 [d] | 9.57 ± 0.5 [d] | 7.20 ± 0.2 [d] | 0.2 ± 0.02 [d] | 3.2 ± 0.6 |
| | PN Skin | 1.4 ± 0.02 | 24.6 ± 0.5 [b] | 21.9 ± 0.2 [b] | 1.2 ± 0.02 [b] | 4.6 ± 0.6 |
| | OA | 0.5 ± 0.02 [c] | 15.9 ± 0.5 [c] | 13.0 ± 0.2 [c] | 0.3 ± 0.02 [c] | 4.0 ± 0.6 |
| | *p*-value * | <0.0001 | <0.0001 | <0.0001 | <0.0001 | 0.19 |
| 4 | Control | 1.7 ± 0.05 [a] | 27.0 ± 0.3 [a] | 25.2 ± 0.3 [a] | 1.5 ± 0.04 [a] | 4.2 ± 0.9 |
| | HO PN | 0.3 ± 0.05 [d] | 8.17 ± 0.3 | 6.69 ± 0.3 [d] | 0.1 ± 0.04 [d] | 3.4 ± 0.9 |
| | PN Skin | 1.4 ± 0.05 [b] | 24.0 ± 0.3 [b] | 21.7 ± 0.3 | 1.2 ± 0.04 [b] | 3.4 ± 0.9 |
| | OA | 0.5 ± 0.05 | 14.5 ± 0.3 [c] | 12.9 ± 0.3 [c] | 0.3 ± 0.04 [c] | 3.2 ± 0.9 |
| | *p*-value * | <0.0001 | <0.0001 | <0.0001 | <0.0001 | 0.68 |
| 6 | Control | 1.5 ± 0.03 [a] | 28.8 ± 0.4 | 23.2 ± 0.4 [a] | 1.4 ± 0.03 [a] | 2.1 ± 0.4 |
| | HO PN | 0.1 ± 0.03 [c] | 8.6 ± 0.4 [d] | 6.4 ± 0.4 [d] | 0.1 ± 0.03 [d] | 2.2 ± 0.4 |
| | PN Skin | 1.2 ± 0.03 [b] | 23.8 ± 0.4 [b] | 21.2 ± 0.4 [b] | 1.2 ± 0.03 [b] | 2.2 ± 0.4 |
| | OA | 0.03 ± 0.03 [d] | 13.7 ± 0.4 [c] | 11.5 ± 0.4 [c] | 0.2 ± 0.03 [c] | 2.5 ± 0.4 |
| | *p*-value * | <0.000 | <0.0001 | <0.0001 | <0.0001 | 0.73 |
| 8 | Control | 1.6 ± 0.05 [a] | 27.7 ± 0.4 [a] | 25.1 ± 0.4 [a] | 1.5 ± 0.04 [a] | 7.1 ± 1.0 |
| | HO PN | 0.2 ± 0.05 [d] | 9.1 ± 0.4 [d] | 6.8 ± 0.4 [d] | 0.1 ± 0.04 [d] | 4.6 ± 1.0 |
| | PN Skin | 1.1 ± 0.05 [b] | 22.5 ± 0.4 [b] | 19.9 ± 0.4 [b] | 1.0 ± 0.04 [b] | 5.9 ± 1.0 |
| | OA | 0.3 ± 0.05 | 14.3 ± 0.4 [c] | 11.9 ± 0.4 [c] | 0.3 ± 0.04 [c] | 5.4 ± 1.0 |
| | *p*-value * | <0.0001 | <0.0001 | <0.0001 | <0.0001 | 0.16 |

Two hundred Hy-Line W36 hens (40-week of lay) were assigned to one of 4 isonitrogenous (18% crude protein) and isocaloric (3080 kcal/kg) treatments (5 replicates per treatment) and fed 8 weeks *ad libitum*. Eggs were collected weekly, and 15 eggs/treatment (3 eggs randomly selected/replicate) N = 60 were chemically analyzed bi-weekly at an AOAC-certified commercial lab, ATC Scientific (Little Rock, AR, USA), using standard AOAC-approved methods. Each value represents the average values over the bi-weekly period ± SEM. Treatments: control = conventional soybean meal and corn mash diet, HOPN = 24% unblanched high-oleic peanut crumbles and corn mash diet, PN Skin = control diet supplemented with 3.0% ground peanut skins, OA = control diet supplemented with 2.5% food-grade oleic fatty acid oil. * *p*-value = differences determined by ANOVA. [a,b,c,d] Means within the same column lacking a common superscript differ significantly ($p < 0.05$).

## 4. Discussion

Numerous feeding trials have demonstrated that the feedstock rations rich in carotenoids (tomato powder, alfalfa, marigold extract) and/or unsaturated fatty acids are transferred to the eggs [10,17]. Studies have also demonstrated improved bioavailability of lutein from enriched eggs in comparison to lutein found in spinach or dietary supplements [18] with enhanced intestinal absorption of lutein when consumed with dietary lipids, suggesting that eggs may be a superior delivery system for some carotenoids. However, the inclusion of specialty feed ingredients (alfalfa meal, marigold, fish meal, linseed meal) is not cost-effective or commercially viable for animal food production. On the contrary, few studies have examined the enrichment of consumable food products (eggs or meat) using agricultural waste by-products rich in polyphenolic compounds such as peanut skins as poultry feedstock ration. Value-addition of agricultural waste by-products, such as peanut skins to poultry feedstock rations, could promote agricultural sustainability and provide creative solutions for agricultural waste by-products with considerable residual nutritional value.

Overall, this study demonstrates that peanut skins can be effectively used in the diets of egg-producing hens at inclusion levels of 3% of the conventional diet, without

adversely affecting hen performance (feed intake, FCR, dozens of eggs produced), egg quality (HU, shell strength, albumen height, vitelline membrane strength, yolk color) or the fatty acid profile of the eggs produced. Additionally, this study parallels previous poultry feeding trials demonstrating that unblanched high-oleic peanuts enrich the eggs and meat produced with unsaturated fatty acids with reduced saturated and trans fats, with reduced egg mass compared to the controls [10] and broiler chickens [19].

While hens fed the HOPN diet had reduced feed intake, 8-week average body weights, and total dozens of eggs produced relative to the other treatments, there were no significant differences in the FCR between the treatment groups. In previous experiments, we demonstrated that hens fed the HOPN dietary treatment had reduced feed intake due to increased ileal fat digestibility and apparent metabolizable energy compared to the other treatment groups [20]. Hence, HOPN fed birds consumed less of a more energetically dense diet to meet the metabolic needs in comparison to the other dietary treatment groups.

In general, egg weights were increased in the PN Skin treatment group relative to the other treatments in the last 4 weeks of the study, while egg weights from the HOPN and OA treatment groups were smaller than the other treatment groups. Similarly, other studies have demonstrated that feeding hens diets rich in unsaturated fatty acids, such as conjugated linoleic acid (CLA), reduce egg weights and body weights [21,22], suggesting that dietary supplementation with CLA causes a reduction in hen body weights, similar to weight loss in humans consuming dietary CLA, which correlates with reduced egg weights and/or size. Egg size has been shown to be greatly influenced by body weight [23]. With every 45 g of body weight increase, there is approximately a 0.5 g increase in egg size from 18 weeks of age in laying hens [24].

In contrast, our previous studies demonstrated that yolk color in eggs produced from hens fed a diet containing unblanched high-oleic peanuts had an approximately 2-fold increase in yolk color in comparison to conventional eggs [10], while in this study, egg yolk color was significantly less in eggs produced from hens fed the HOPN diet relative to the other treatment groups. Eggs produced from hens fed the PN Skin and control diets had similar levels of palmitic and elaidic fatty acid for most of the study, while eggs produced from the HOPN and OA treatment groups had reduced saturated and trans fatty acid levels comparatively.

Chemical analysis of the four experimental diets revealed increased levels of omega 3 fatty acid levels in the PN Skin diet relative to the other dietary treatments. Nonetheless, omega 3 fatty acid level in eggs produced from hens fed the PN Skin experimental diet was similar between all treatment groups. The soybean oil, whole peanuts and/or peanut skins, and yellow corn (very low levels) are the predominate feed ingredients containing omega 3 fatty acids [25–27], which may have correlated to elevated omega 3 fatty acid content found in the PN Skin experimental diets that contained modest amounts of each of these feed ingredients relative to the other dietary treatments.

In contrast to our previously published reports [10], β-carotene content in this study was not significantly different between eggs produced from hens fed the four different dietary treatment groups at any of the experimental time points measured in this study. Studies conducted by Pattee and Purcell (1967) revealed that peanut oil extracted from young peanuts contained 60 μg of β-carotene and 138 μg of lutein per liter, while peanut oil extracted from more mature peanuts had lower concentrations of these carotenoids [28]. However, the determination of peanut maturity has been correlated with the increasing color of the mesocarp from white to yellow, orange, brown, and black [29]. Peanuts have an indeterminate growth pattern, in which at harvest, the combine collects peanut pods ranging in different maturity levels present on the plant [30]. Therefore, a given peanut harvest may contain a higher percentage of young/immature pods that contain elevated levels of carotenoids in the seed and oil [28], suggesting that a potentially higher percentage of young/immature peanuts may have been harvested for use in our earlier layer hen feeding trials with unblanched high-oleic peanuts in which enriched the eggs produced with unsaturated fatty acids and β-carotene [10].

β-carotene is a lipid-soluble carotenoid found abundantly in plants and responsible for the rich yellow and deep orange colors in plants [31]. Conventional commercial eggs are rich in lutein and zeaxanthin [32], which are carotenoids that are most likely transferred from yellow corn in the diet to the egg yolks. Layer feeding trials have demonstrated the transfer of carotenoids and their pigments from the diet to the yolks of eggs produced [9]. While our previous layer feeding trials demonstrated that eggs produced from hens fed a HOPN diet had significantly increased β-carotene levels and yolk color relative to conventional eggs, in which the rich yellow/deep orange pigment of β-carotene was transferred to the eggs. However, in this study, β-carotene content was not elevated, and hence the yellow/orange pigments were not available to transfer from the diet to the yolks of eggs produced by hens fed the HOPN or PN Skin diets. Moreover, eggs produced from hens fed the HOPN diet had less available dietary carotenoids from yellow corn (lutein and zeaxanthin) in the diet relative to the other treatment groups (Table 1 content of yellow corn: control 47.5%, 36.9%, PN Skin 56.9%, OA 57.0%), which may have also correlated to reduced yolk color scores.

Most importantly, this study reports similar body weights, feed intake, FCR, and egg chemistry between the PN Skin and control treatment groups, implying the effective use of PN skins as an alternative layer feed ingredient. These results support the value-added use of peanut skins as a poultry feed ingredient, an abundant agricultural waste by-product of the peanut industry. While this study has positive implications for the use of peanut skins as an alternative poultry feed ingredient, this study fails to parallel commercial egg production commonly using floor pens or alternative housing systems. Moreover, we aim to repeat this study with larger sample sizes to more closely parallel industry. In addition, we aim to conduct additional feeding trials with hens housed in floor pens and fed a peanut and/or peanut skin-containing diet for comparative analysis of the production performance to hens housed in battery cages and fed a peanut and/or peanut skin-containing diet.

**Author Contributions:** All authors actively contributed to the care and husbandry of all research animals, while co-authors O.T., R.M. and K.A. were active participants in the data analysis, data interpretation, and preparation of the manuscript. All authors have read and agreed to the published version of the manuscript.

**Funding:** This work was funded by the Food Science & Market Quality and Handling Research Unit, Agricultural Research Service (CRIS Project 6070-43440-012-00D), and the North Carolina Peanut Growers Association (Award (1119) 2021-0561).

**Institutional Review Board Statement:** The procedures used in these studies were approved by the North Carolina State University Institutional Animal Care and Use Committee (IACUC #17-001-A).

**Informed Consent Statement:** Informed consent was obtained from all subjects involved in the study. The sensory protocol was reviewed and deemed exempt by the NC State University Institutional Review Board (IRB) for human subjects.

**Data Availability Statement:** The data presented in this study are available on request from the corresponding author.

**Acknowledgments:** The authors would like to acknowledge the students and staff of the Prestage Department of Poultry Science, Hampton Farms-Jimbo's Jumbos and to Birdsong Peanuts for a donation of high-oleic peanuts for this feeding trial.

**Conflicts of Interest:** The authors declare no conflict of interest.

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
