# Peer review of "The Effect of Feeding Hens a Peanut Skin-Containing Diet on Hen Performance, and Shell Egg Quality and Lipid Chemistry"

_agriculture, doi:10.3390/agriculture11090894_

Round 1
Reviewer 1 Report
Generally, it is a well-written article on an interesting and up-to-date subject. Please find my comments regarding the manuscript.
In the abstract, we can find that “Two hundred commercial hens were randomly assigned to 4 treatments (20 replicates)”, but in the Materials and Methods, it was stated that “…were randomly assigned to one of 4 … treatments, with 5 replicates per treatment.”…. So which version is the right one?
Line 19, in the abstract, is “≤” and should be “<” or generally I suggest removing all info about sign. level form the abstract, like (p<0.05) or (p<0,0001) etc., since we can find the information about significance level in the 2.4 Statistical Analysis part.
Line 297, Table.3, p-value are shifted and do not match the columns. Generally; the Discussion part needs to be improved by adding real discussion while now in many cases it contains the only description of the results, for example: - Authors mentioned differences in the weight of eggs but did not give any explanation of that fact,- Authors pointed out that in this study egg yolk colour was sign. less in eggs from hens of the HOPN group while in previous studies it was the other way,- Authors stated that the omega3 level in the PN Skin diet was higher than in other treatments, incl. control (58.5 vs 6.6 in control) but during the all experimental period, the content of omega3 was the highest in the eggs from the control group, WHY???
Author Response
Agriculture (ISSN 2077-0472)
Manuscript ID
agriculture-1359341
The Effect of Feeding Hens a Peanut Skin-Containing Diet on Hen Performance, and Shell Egg Quality and Lipid Chemistry
Ondulla Toomer * , Thien Vu , Rebecca Wysocky , Vera Moraes , Ramon Malheiros , Kenneth Anderson
Reviewer 1
Generally, it is a well-written article on an interesting and up-to-date subject. Please find my comments regarding the manuscript.
Author Response:
Thank you!
Reviewer 1
In the abstract, we can find that “Two hundred commercial hens were randomly assigned to 4 treatments (20 replicates)”, but in the Materials and Methods, it was stated that “…were randomly assigned to one of 4 … treatments, with 5 replicates per treatment.”…. So which version is the right one?
Author Response:
In the abstract this typo was corrected to 5 replicates in the abstract. The study was conducted with 200 hens with 4 treatments and 5 replicates per treatment with 10 hens per replicate. Please see the edited manuscript.
Line 19, in the abstract, is “≤” and should be “<” or generally I suggest removing all info about sign. level form the abstract, like (p<0.05) or (p<0,0001) etc., since we can find the information about significance level in the 2.4 Statistical Analysis part.
Author Response:
In the abstract the p-values were removed from the abstract per the reviewers’ comments. Please see the edited manuscript.
Line 297, Table.3, p-value are shifted and do not match the columns.
Author Response:
In table 3 of the edited manuscript the p-values were re-located under the correct columns. Please see the edited manuscript.
Generally; the Discussion part needs to be improved by adding real discussion while now in many cases it contains the only description of the results, for example: -
Authors mentioned differences in the weight of eggs but did not give any explanation of that fact,-
Author Response:
Per the reviewers’ comments explanations have been added in the “discussion” section of the manuscript. Please see lines 492 to 501 of the edited manuscript.
Authors pointed out that in this study egg yolk colour was sign. less in eggs from hens of the HOPN group while in previous studies it was the other way,-
Author Response:
Per the reviewers’ comments explanations have been added in the “discussion” section of the manuscript. Please see lines 528 to 541 of the edited manuscript.
Authors stated that the omega3 level in the PN Skin diet was higher than in other treatments, incl. control (58.5 vs 6.6 in control) but during the all experimental period, the content of omega3 was the highest in the eggs from the control group, WHY???
Author Response:
Per the reviewers’ comments explanations have been added in the “discussion” section of the manuscript in lines 512 to 517, stating the following: “The soybean oil, whole peanuts and/or peanut skins and yellow corn (very low levels) are the predominate feed ingredients containing omega 3 fatty acids [25, 26, 27], which may have correlated to elevated omega 3 fatty acid content found in the PN Skin experimental diets that contained modest amounts of each of these feed ingredients relative to the other dietary treatments.” Please see the revised manuscript.

Reviewer 2 Report
The manuscript “The Effect of Feeding Hens a Peanut Skin-Containing Diet on Hen Performance, and Shell Egg Quality and Lipid Chemistry” reports data from an interesting study on the use of the waste product of food industry, thus supporting the sustainability of egg production. The described study is therefore relevant to the readers of the journal.
The study seems well executed, and the manuscript is fluently written. I have only minor comments on the manuscript. I would like the authors’ clarification regarding a few points
- The study was conducted in battery cages. The authors could state reasons for this. Nowadays, in many parts of the world, egg production in battery cage system is getting rarer and is replaced with different kinds of floor systems. Do the authors expect that the results would be similar in all kinds of production systems?
- Discussion
- a paragraph focusing on the strengths and limitations would be appreciated. Also, the authors could discuss if their results revealed the need for further studies.
- in the last paragraph, concluding that the results of this study only support the results of previous studies seem to lower the value of the results. In conclusions, the authors should emphasize the new information this particular study provides.
Author Response
Agriculture (ISSN 2077-0472)
Manuscript ID
agriculture-1359341
The Effect of Feeding Hens a Peanut Skin-Containing Diet on Hen Performance, and Shell Egg Quality and Lipid Chemistry
Ondulla Toomer * , Thien Vu , Rebecca Wysocky , Vera Moraes , Ramon Malheiros , Kenneth Anderson
Reviewer 2
The manuscript “The Effect of Feeding Hens a Peanut Skin-Containing Diet on Hen Performance, and Shell Egg Quality and Lipid Chemistry” reports data from an interesting study on the use of the waste product of food industry, thus supporting the sustainability of egg production. The described study is therefore relevant to the readers of the journal.
The study seems well executed, and the manuscript is fluently written. I have only minor comments on the manuscript. I would like the authors’ clarification regarding a few points
Author Response: Thank you
- The study was conducted in battery cages. The authors could state reasons for this. Nowadays, in many parts of the world, egg production in battery cage system is getting rarer and is replaced with different kinds of floor systems. Do the authors expect that the results would be similar in all kinds of production systems?
- Author Response: Due to animal welfare concerns, many poultry and egg producers have transitioned from battery cage housing to alternative housing (alternative enhanced caging systems, free-range systems and/or floor pens). Thus, it is in our future to conduct additional peanut/peanut skin layer feeding trials with hens housed in floor pens for comparative analysis of production performance of hens housed in battery cages versus floor pens and fed similar peanut/peanut skin-containing diets.
Per the reviewers’ comments this comment has been added to line 551 to 554 of the edited manuscript.
- Discussion
- a paragraph focusing on the strengths and limitations would be appreciated. Also, the authors could discuss if their results revealed the need for further studies.
- Author Response: In response to the reviewers’ comments Line 558 to 564 were added to the body of the text in the discussion section detailing the limitations of this study. Which include the following statements in the edited manuscript: “ While this study has positive implications for the use of peanut skins as an alternative poultry feed ingredient, this study fails to parallel commercial egg production commonly utilizing floor pens or alternative housing systems. Moreover, we aim to repeat this study with larger sample sizes to more closely parallel industry. Also, we aim to conduct additional feeding trials with hens housed in floor pens and fed a peanut and/or peanut skin-containing diet for comparative analysis of the production performance to hens housed in battery cages and fed a peanut and/or peanut skin-containing diet.”
- in the last paragraph, concluding that the results of this study only support the results of previous studies seem to lower the value of the results. In conclusions, the authors should emphasize the new information this particular study provides.
Author Response: In response to the reviewers’ comments Line 553 to 557 were added in the last 2 conclusive paragraphs detailing the importance of this study, in the following statements of the edited manuscript: “Most importantly, this study reports similar body weights, feed intake, FCR and egg chemistry between the PN Skin and control treatment groups, implying the effective utilization of PN skins as an alternative layer feed ingredient. These results support the value-added utilization of peanut skins as a poultry feed ingredient, an abundant agricultural waste by-product of the peanut industry.”
